# DynamicBind: predicting ligand-specific protein-ligand complex structure with a deep equivariant generative model

Wei Lu [1,5] ✉, Jixian Zhang[1,5] ✉, Weifeng Huang[2], Ziqiao Zhang[1], Xiangyu Jia[1], Zhenyu Wang[1], Leilei Shi [1], Chengtao Li[1], Peter G. Wolynes[3] & Shuangjia Zheng [4,5] ✉

While significant advances have been made in predicting static protein structures, the inherent dynamics of proteins, modulated by ligands, are crucial for understanding protein function and facilitating drug discovery. Traditional docking methods, frequently used in studying protein-ligand interactions, typically treat proteins as rigid. While molecular dynamics simulations can propose appropriate protein conformations, they're computationally demanding due to rare transitions between biologically relevant equilibrium states. In this study, we present DynamicBind, a deep learning method that employs equivariant geometric diffusion networks to construct a smooth energy landscape, promoting efficient transitions between different equilibrium states. DynamicBind accurately recovers ligand-specific conformations from unbound protein structures without the need for holo-structures or extensive sampling. Remarkably, it demonstrates state-of-the-art performance in docking and virtual screening benchmarks. Our experiments reveal that DynamicBind can accommodate a wide range of large protein conformational changes and identify cryptic pockets in unseen protein targets. As a result, DynamicBind shows potential in accelerating the development of small molecules for previously undruggable targets and expanding the horizons of computational drug discovery.

Remarkable progress has been achieved in the realm of protein structure prediction from sequence data. Some prediction techniques use machine learning in concert with molecular dynamics or Monte Carlo[1–4]. These generate an ensemble of structures. AlphaFold, which leads the way in the prediction of nearly all structures in the human proteome[5–8], however, typically generates only a few conformations for each protein sequence, despite the fact that proteins are inherently dynamic and generally adopt multiple conformations to perform their functions[9,10]. The ability of proteins to interconvert between different conformations is central to their biological activities in all domains of life. The therapeutic effect of drug molecules arises from their specific binding to only some conformations of the target proteins and thereby modulating essential biological activities by altering the conformational landscape of these proteins[11–14]. In practice, nowadays the interactions between proteins and ligands are studied through molecular docking methods computationally. Docking is a key component of structure-based drug discovery[15]. Nevertheless, despite the widespread recognition of the importance of protein dynamics, traditional

[1]Galixir Technologies, 200100 Shanghai, China. [2]School of Pharmaceutical Science, Sun Yat-sen University, 510006 Guangzhou, China. [3]Center for Theoretical Biological Physics and Department of Chemistry, Rice University, Houston, TX 77005, USA. [4]Global Institute of Future Technology, Shanghai Jiao Tong University, 200240 Shanghai, China. [5]These authors contributed equally: Wei Lu, Jixian Zhang, Shuangjia Zheng. ✉e-mail: luwei0917@gmail.com; jxzly1993@gmail.com; shuangjia.zheng@sjtu.edu.cn

docking methods often treat proteins as rigid, or in some cases, as being only partially flexible, permitting only selected side chains to move, to manage computational costs[16,17]. As a result, AlphaFold-predicted structures of apoproteins, when used as inputs for docking, will yield ligand pose predictions that do not align well with the ligand-bound co-crystallized holo-structures[18,19]. The AlphaFold-predicted structures often do not present the most favorable side-chain rotamer configurations for ligand binding, and consequently the relevant binding pocket will appear to be inaccessible since the apoprotein adopts a conformation substantially different from the holo state.

Here, we present DynamicBind, a geometric deep generative model designed for "dynamic docking". Unlike traditional docking methods that treat proteins as mostly rigid entities, DynamicBind efficiently adjusts the protein conformation from its initial AlphaFold prediction to a holo-like state. Our model is capable of handling a wide range of large conformational changes during prediction, such as the well-known DFG-in to DFG-out transition in kinase proteins, a challenge that has been formidable for other methods, such as molecular dynamics (MD) simulations[20,21]. We have attained this efficiency in sampling large protein conformational changes by learning a funneled energy landscape, where the transitions between biologically relevant states are minimally frustrated[22]. This is made possible through the innovative employment of a morph-like transformation for decoy generation during training (more in "DynamicBind architectures" and "Methods"). The present method shares similarities with the Boltzmann generator[23,24], as it allows for direct and efficient sampling of low-energy states from the learned model. Unlike traditional Boltzmann generators, which are typically constrained to the systems for which they are trained on, however, DynamicBind is a generalizable model that can handle new proteins and ligands.

In the upcoming results section, we present a comprehensive evaluation of DynamicBind, illustrating its potential to aid drug discovery. Our presentation is organized into six segments: first, outlining the DynamicBind model; then benchmarking DynamicBind against current docking methods; then highlighting the method's ability to sample large protein conformational changes in a ligand-specific manner; then specifically demonstrating the scope of conformational changes it can handle; illustrating its capacity to predict cryptic pockets through a case study; and finally showcasing its application to the proteome-wide virtual screening task using an antibiotics dataset[25]. These investigations collectively highlight the potential of DynamicBind, setting the stage for further understanding and manipulating the protein–ligand interaction landscape.

## Results

### DynamicBind architectures

DynamicBind executes "dynamic docking", a process that performs prediction of the protein–ligand complex structure while accommodating substantial protein conformational changes. DynamicBind accepts apo-like structures (in the present study, AlphaFold-predicted conformations) in PDB format and small-molecule ligands in several widely available formats, such as Simplified Molecular Input Line Entry System (SMILES) or structure-data file (SDF) format. During inference, the model randomly places the ligand, whose seed conformation is generated using RDKit[26], around the protein. Then, over the course of 20 iterations (more details in "Model architecture"), using progressively smaller time steps, the model gradually translates and rotates the ligand while adjusting its internal torsional angles. After the initial five steps where only the ligand conformation is changed, the model then simultaneously translates and rotates the protein residues, while modifying the side-chain chi angles[27], in the remaining steps.

As illustrated in Fig. 1a, at each step, the features and the coordinates of the protein and the ligand are fed into an SE(3)-equivariant interaction module. Subsequently, the protein and readout modules generate the predicted translation, rotation, and dihedral updates for

the current state. Further details about the model are given in "Transformation of the protein conformation". Unlike the traditional protocol employed in diffusion-based model training, which generates decoys by perturbing the native state with Gaussian noise of varying magnitudes[28–32], our method employs a morph-like transformation to produce protein decoys. In the process of doing this, the native conformation is gradually transitioned towards the AlphaFold-predicted conformation. The structure of proteins is highly constrained in many ways, with residues linked by peptide bonds, and the bond lengths are governed by chemical principles. When decoys are generated using Gaussian noise, the model primarily learns only to revert to the most chemically stable conformation, often the conformation before the noise was added. In the present task, the ligand-bound holo conformation is unknown, and the most readily available protein structure is the one predicted by AlphaFold, which often significantly differs from the holo conformation. Given that the AlphaFold-predicted structure often already complies with most chemical constraints, it is challenging to anticipate how the model trained on decoys made merely from Gaussian noise could accurately predict long timescale transformations of biological relevance, which are our primary concern. In contrast, the decoys generated by our morph-like transformation generally satisfy the basic chemical constraints, allowing our model to concentrate on learning biophysically relevant state-changing events. In unbiased molecular dynamics simulations, transitions between meta-stable states, such as the DFG 'in' and 'out' transition, are infrequent due to the realistic yet rugged energy landscape inherent in the all-atom force field[21]. Our method, in contrast, features a significantly more funneled energy landscape, effectively lowering the free energy barrier between biologically meaningful states. Consequently, akin to other Boltzmann generator methods[23,33], the present approach demonstrates markedly enhanced efficiency in sampling alternate states pertinent to ligand binding. A schematic figure has been included to elucidate these differences (Fig. 1b).

### DynamicBind achieves higher accuracy in ligand pose prediction and improves the initial AlphaFold-predicted protein conformations

To evaluate our method, we first utilized the PDBbind dataset[34] and, in line with previous works[19,35,36], we trained the model using a chronological, time-based split of the training, validation, and test sets. Since the PDBbind test set, comprising around 300 structures from 2019, includes many non-small-molecule ligands (53 cases being polypeptides), we extended the scope of our assessment using a curated Major Drug Target (MDT) test set. The MDT set includes 599 structures that were deposited in or after 2020, with both drug-like ligands and proteins from four major protein families: kinases, GPCRs, nuclear receptors, and ion channels (refer to "Dataset construction" for more details). These protein families represent the targets of about 70% of FDA-approved small-molecule drugs[37].

Instead of using holo-structures as the input, we adopted a more challenging and realistic scenario during testing, where we assumed the holo protein conformation is not available and only use the protein conformations predicted by AlphaFold as our input. Holo conformations exhibit strong shape and charge complementarity to co-crystallized ligands, which already, unrealistically, simplify ligand pose prediction[11]. In contrast, the apo conformations or those predicted by AlphaFold may clash with transplanted ligands obtained by superimposing crystal structures[14].

As shown in Fig. 2a and b, DynamicBind predicts more cases with ligand RMSD below various thresholds than other baselines. In particular, it achieves the fraction of ligand RMSD below 2 Å (5 Å), being 33% (65%) on the PDBbind test set and 39% (68%) on the MDT test set, respectively.

Evaluating models solely on ligand RMSD may favor deep learning-based models (DiffDock, TankBind, and DynamicBind) due to

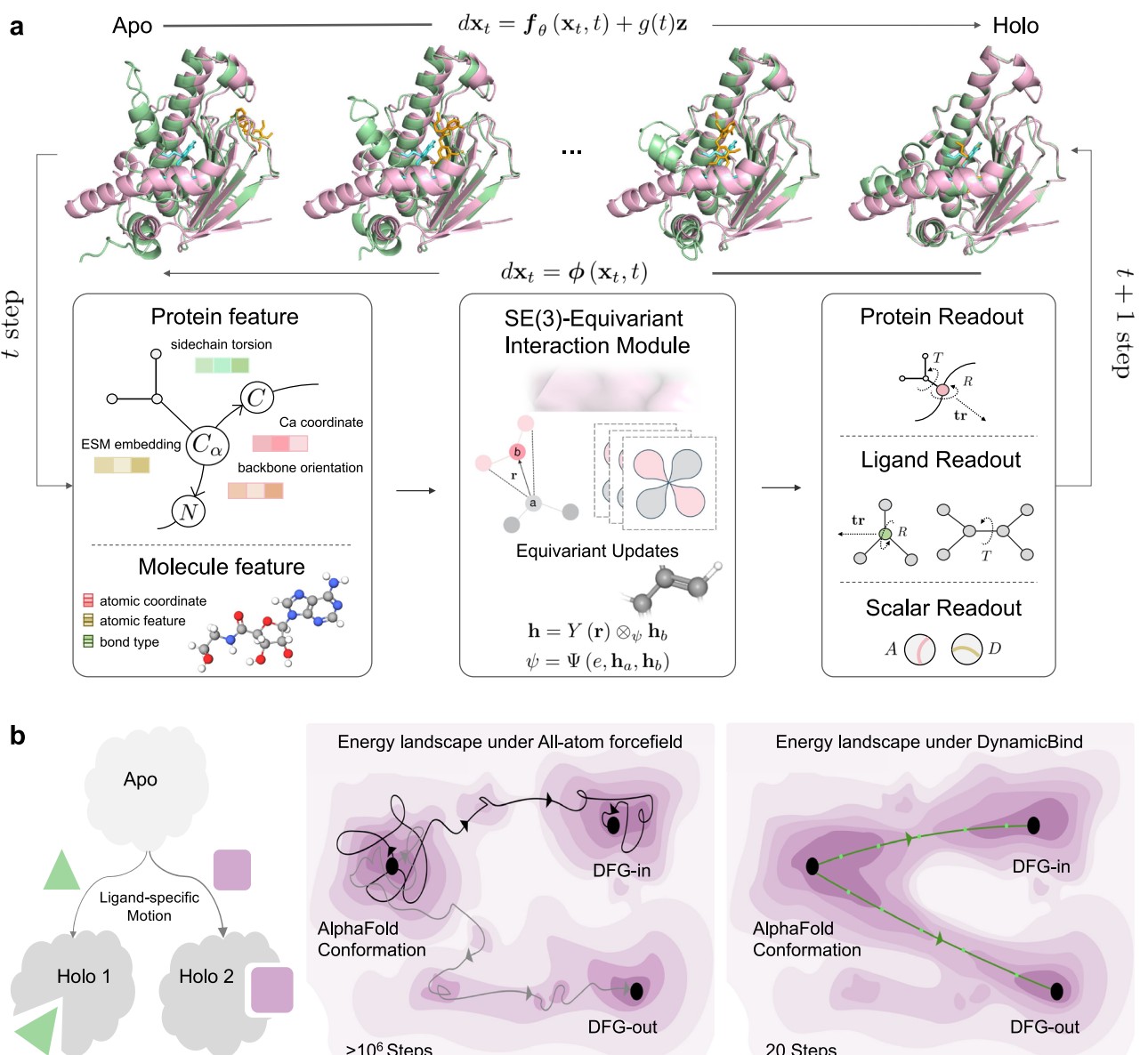

**Fig. 1 | Overview of DynamicBind model. a** The holo state is represented in pink, the initial apo and the model-predicted conformation in green. The native ligand is depicted in cyan, and the predicted ligand shown in orange. The model accepts as input both the features and the current conformation of the protein and ligand. The output readouts include the predicted updates: global translation and rotation for both the ligand and each protein residue, the rotation of torsional angles for the ligands and chi angles for the protein residues, and two prediction modules (binding affinity, A and confidence score, D). During the training phase, the model is designed to learn the transformation from the apo-like conformation into the holo conformation. During inference, the model iteratively updates the initial input structure twenty times. **b** A schematic figure shows that our model could predict the two different holo conformations when the protein binds with two different ligands. Our model could predict the bounded protein conformation within 20 steps, while millions of steps of all-atom MD simulations are needed to find the same bounded state.

their higher clash tolerance, while may disadvantage force field-based methods (GNINA, GLIDE, VINA) that strictly enforce Van der Waals forces. Significant clashes can impede interaction analysis in structure-based drug design, obscuring crucial molecular interactions and complicating the design of molecule improvements. Consequently, we use both ligand RMSD and clash scores (as defined by Hekkelman et al.[14]) to assess success rates. Figure 2c, shows the success rates using both a stringent criterion, ligand RMSD < 2 Å, clash score <0.35, and a more relaxed criterion, ligand RMSD < 5 Å, clash score < 0.5. The success rate of DynamicBind (0.33) is 1.7 times higher than the best baseline DiffDock (0.19) under the more stringent condition. Furthermore, DynamicBind has demonstrated the ability to reduce the pocket RMSD relative to the initial AlphaFold structure, even in cases with large original pocket RMSDs (Fig. 2d). This observation highlights that the present approach is capable of managing substantial conformational changes, recovering holo-structures when other methods may struggle. Given our model's ability to generate diverse conformations, we developed the contact-LDDT (cLDDT) scoring module, a concept inspired by AlphaFold's LDDT score. The module's purpose is to select the most suitable complex structure from the predicted outputs. As shown in Fig. 2e, our predicted cLDDT correlates well with the actual ligand RMSD, indicating its effectiveness in selecting high-quality complex structures. The auROC score, with ligand RMSD below 2 Å as the true positive, is 0.764. While our cLDDT scoring function is effective, there is potential for improvement. Perfect selection could enhance our success rate from 0.33 to 0.5, as illustrated in Fig. 2f. Even

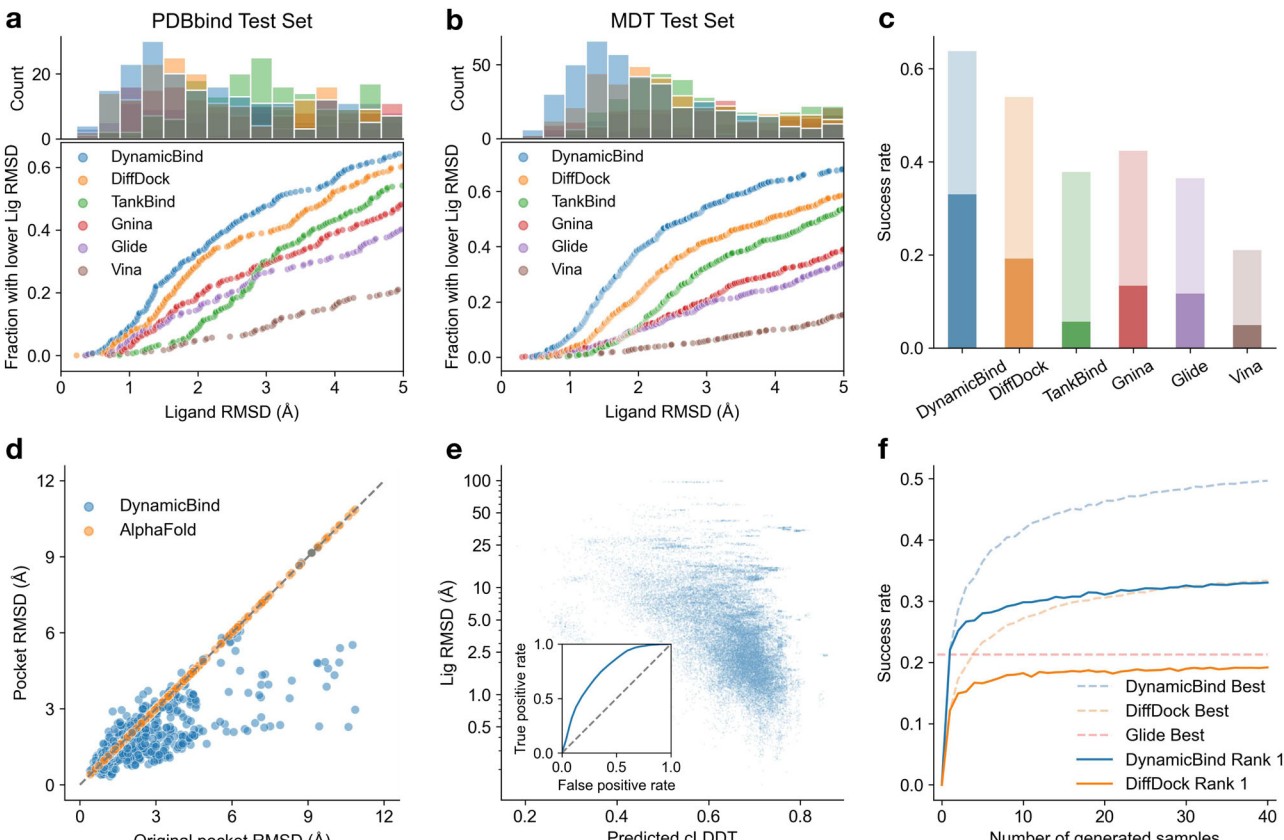

**Fig. 2 | Benchmark results overview. a**, **b** DynamicBind outperforms other methods in predicting ligand poses for both the PDBbind dataset (**a**) and major drug targets (MDT) dataset (**b**) across different RMSD thresholds. **c** Dark and light shades represent success rates under stringent (ligand RMSD < 2 Å, clash score < 0.35) and relaxed (ligand RMSD < 5 Å, clash score < 0.5) criteria, respectively. **d** The protein conformations predicted by DynamicBind are more native-like, as evidenced by the lower pocket RMSD around the binding sites. **e** The contact-LDDT (cLDDT) score predicted by DynamicBind correlates well with the ligand RMSD and is a good predictor of the true ligand RMSD below 2 Å (auROC 0.764). **f** As the number of generated samples increases, the success rate increases. **c**–**f** display results for the combined PDBbind and MDT test sets. Results for individual datasets, as well as those filtered at 30%, 60%, and 90% maximum ligand and protein sequence similarity cutoffs, are detailed in Supplementary Figs. 1–4. Source data are provided as a Source Data file.

in the absence of an ideal selection model, our method considerably outperforms DiffDock and the top force field-based method, GLIDE. Due to the variability in the number of samples produced by Glide, often because of its filtering scheme eliminating unrealistic conformations, Glide's best performance is represented using a flat line. This line reflects the success rate determined by the most effective sample from Glide. DynamicBind's exceptional performance stems from its ability to undergo significant protein conformational changes, leading to a better fit between the protein and the ligand.

To assess the model's generalization to new proteins and ligands, we analyzed results stratified by maximum ligand and protein sequence similarity to the training set (Supplementary Figs. 3 and 4). This analysis reveals that DynamicBind performs well with new ligands, outperforming others, but is less effective with new proteins, where it is outperformed by classical docking methods with predefined ground-truth binding pockets. Other deep learning methods also show similar declines with new proteins, hinting at a need for larger training set and improved inductive biases. Moreover, considering that the identification of binding sites on new proteins is an active research area, the challenges encountered in blind global docking by deep learning methods, including ours, are likely shared across different approaches. Overall, DynamicBind's proficiency with new ligands is significant in drug discovery, highlighting its potential in identifying protein conformational changes vital for creating effective, specific drugs.

## DynamicBind can capture ligand-specific protein conformational changes

Conventional docking protocols usually perform protein conformation sampling as a separate step from the docking process[15,38]. In many instances, however, two distinct ligands may fit into mutually exclusive protein conformations. For example, c-Met kinase can adopt two different conformations, corresponding to active and inactive states, typically referred to as the Asp–Phe–Gly (DFG)-in and DFG-out conformations (Fig. 3b, d). The DFG motif can flip out, subsequently blocking or opening up different regions of the protein. In previous docking models, the protein must be preset to the correct conformation to have a chance of identifying the appropriate binding pose for the ligand[20]. In contrast, DynamicBind, utilizing the protein conformation predicted by AlphaFold (Fig. 3a), can dynamically adjust the protein conformation to find the optimal conformation that accommodates the ligand of interest. As a representative case, for PDB 6UBW, the predicted ligand RMSD is 0.49 Å, and pocket RMSD is 1.97 Å, while the pocket RMSD for the AlphaFold structure is 9.44 Å. For PDB 7V3S, the predicted ligand RMSD is 0.51 Å, and the pocket RMSD is 1.19 Å, (AlphaFold 6.02 Å). Neither of the two ligands have been seen before in the training set (Fig. 3c, e). In our quantitative analysis, only seven proteins from the test set, represented in 79 PDB structures, were found to adopt both DFG-in and DFG-out conformations, as annotated by the Kinase–Ligand Interaction Fingerprints and Structures (KLIFS) web server[39]. Figure 3f and g demonstrates how

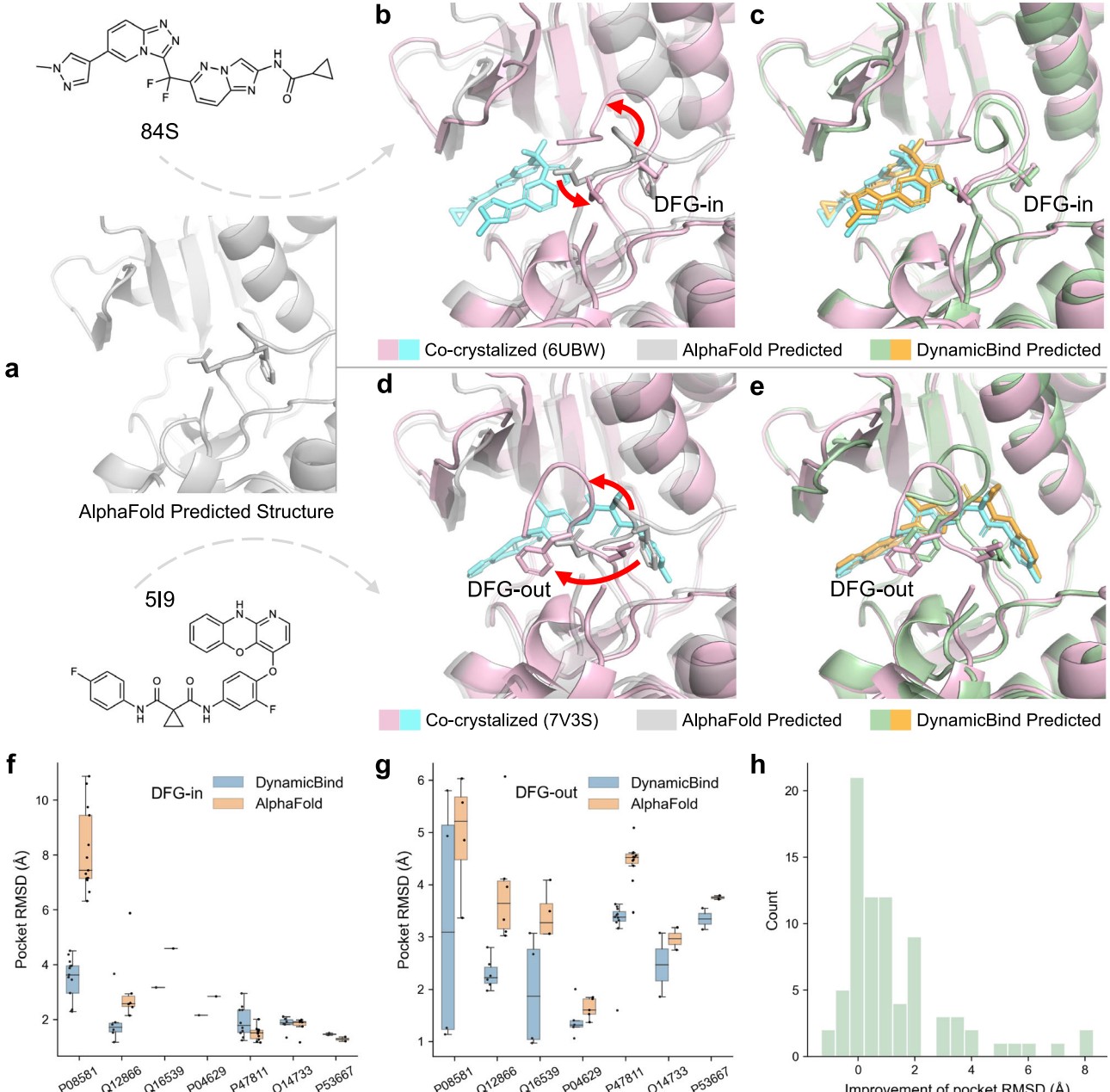

**Fig. 3 | DynamicBind captures ligand-specific protein conformational changes.** AlphaFold-predicted structures are depicted in white, the crystal structure with protein, and ligand in pink and cyan, respectively. Our model's predictions are shown in green and orange, for the protein and ligand, respectively. The side chains of the Asp–Phe–Gly (DFG) residues are shown in stick. Red arrows highlight significant conformational changes of the crystal structure from the AlphaFold structure. The input conformation is the AlphaFold-predicted conformation. **a** When the ligand 84S (**b**) binds to c-Met protein, the protein adopts a DFG-in conformation. When the ligand 5I9 (**d**) binds to the same protein, the protein adopts a DFG-out conformation. Our prediction for both ligands (**c**, **e**) agrees well with the crystal structure. Ligand RMSD is 0.49 Å and 0.51 Å. Improvement of Pocket RMSD from initial AlphaFold is 7.47 Å and 4.83 Å for DFG-in and DFG-out, respectively. Among the test set, seven proteins (identified by their UniProt IDs), contains both DFG-in and DFG-out crystallized holo conformations, their pocket RMSD of both initial AlphaFold and predicted structures are shown in (**f**, $n = 39$) and (**g**, $n = 34$) for DFG-in holo conformations and DFG-out holo conformations separately, where central line marking the median, box edges indicating the upper and lower quartiles, whiskers extending up to 1.5 times the interquartile range, and individual points in dots. **h** The histogram of the improvement in pocket RMSD from AlphaFold for all 79 PDBs. Source data are provided as a Source Data file.

these proteins (denoted by their UniProt IDs), starting from the same initial structure, move progressively towards the DFG-in conformation upon type-I inhibitor binding, and incline towards the DFG-out conformation when interacting with a type-II inhibitor. Further, Fig. 3h reveals that the majority of the predicted protein structures show a lower pocket RMSD compared to the initial AlphaFold structures. These results demonstrate that, DynamicBind, is capable of capturing ligand-specific conformational changes. This feature is critical in

preventing the overlooking of potential "hit" compounds that could bind well with conformations distinct from the initially provided protein structure.

**DynamicBind covers multi-scale protein conformation changes**
The DFG-in/out conformation has been extensively studied, and some challenges can be partially addressed by employing ensemble docking, wherein proteins in both conformations are utilized for docking[40,41].

Ensemble docking, however, elevates computational costs and may not be suitable for less well-characterized conformations. In this section, we provide a comprehensive analysis of six distinct conformational changes across the picosecond level to millisecond level, each exemplified by a case found in our PDBbind test set. In Fig. 4, the crystal structure is depicted in pink, the AlphaFold structure in white, and our prediction in green. The native ligand is illustrated in cyan, and our predicted ligand is in orange. Δpocket RMSD measures the difference in pocket RMSD between the predicted protein structure and the AlphaFold structure, based on comparison with the crystal structure. A negative Δpocket RMSD indicates that the predicted aligns more closely with the crystal structure compared with the AlphaFold prediction. Δclash measures the difference in clash scores between the predicted protein–ligand pair and the AlphaFold structure with the transplanted ligand[14]. A negative Δclash indicates fewer clashes in the predicted complex. In Fig. 4a, the native ligand clashes with a side chain of the superimposed AlphaFold structure; in our prediction, this side chain shifts towards the native conformation, thus resolving the clash. In Fig. 4b, a part of the pocket is blocked by a Tyrosine in the AlphaFold structure; it becomes accessible in both our predicted and native structures. In Fig. 4c, a flexible loop intersects with the ligand, and it moves away in our prediction, consistent with the native structure. In Fig. 4d, alpha helices transform into loops near the ligand-binding site. In Fig. 4e, a substantial secondary structure motion is observed in the Heat shock protein, Hsp90α, transitioning from the closed state to the open state. In Fig. 4f, two domains of AKT1 kinase coalesce, forming a pocket that did not previously exist. Taken together, the present model can predict diverse types of conformational changes associated with ligand binding when the ligand-binding pocket is either insufficiently spacious or unformed in the AlphaFold-predicted conformations.

## DynamicBind reveals cryptic pockets significant to drug discovery

The dynamic nature of proteins often gives rise to cryptic pockets. These cryptic pockets, which appear during protein dynamics, can reveal druggable sites not found in static structures, thus making previously 'undruggable' proteins into potential drug targets. We demonstrate the utility of DynamicBind in revealing these cryptic pockets using the SET domain-containing protein 2 (SETD2), a histone methyltransferase, as a case study. SETD2, critical for the treatment of multiple myeloma (MM) and diffuse large B-cell lymphoma (DLBCL)[42,43], has a cryptic pocket targeted by a highly selective compound, EZM0414, currently undergoing Phase I clinical trials. As illustrated in Fig. 5a, b, all SETD2 homologs in the training set, defined by a protein Smith–Waterman similarity[44] over 0.4, are co-crystallized with S-Adenosyl methionine (SAM) or Sinefungin analogs, depicted in lines. Sinefungin and its analogs broadly inhibit methyltransferases by occupying the SAM site[45], making the selective inhibition of SETD2 challenging. Before 2019, no structure of SETD2 or its homologs had

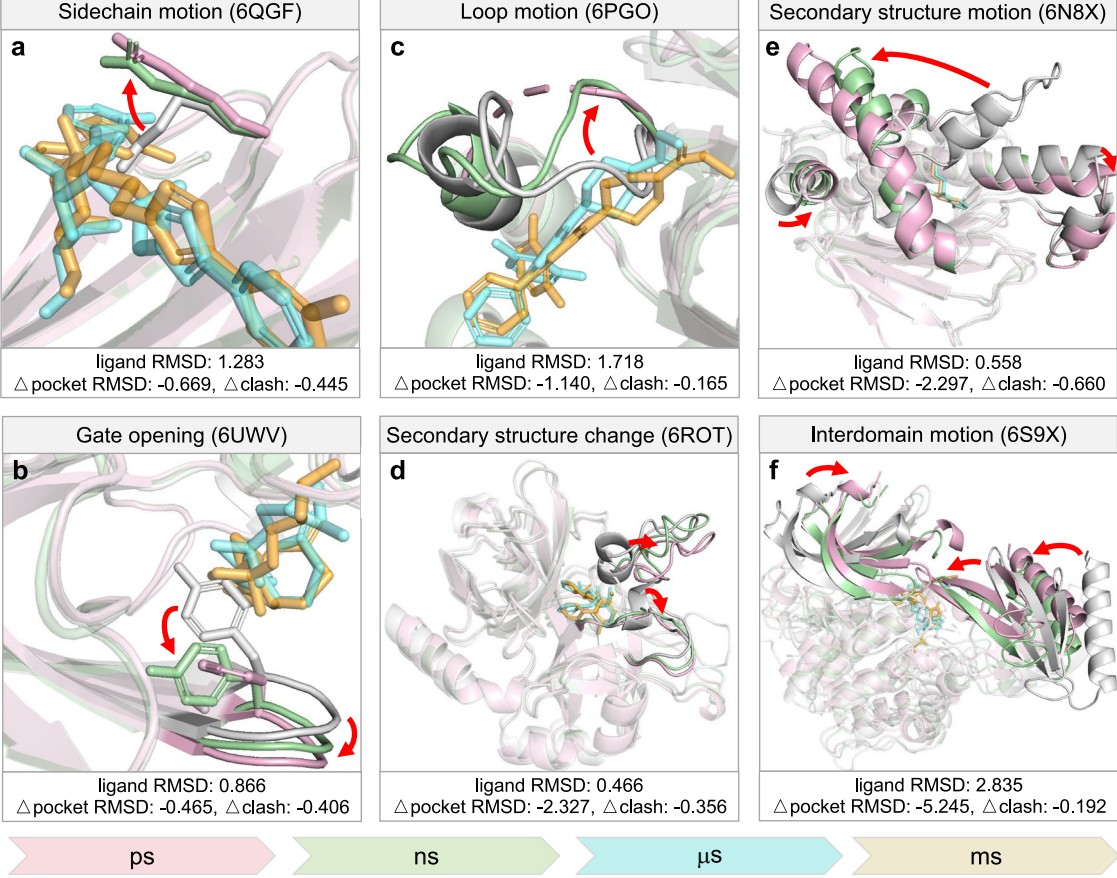

| Sidechain motion (6QGF) | Loop motion (6PGO) | Secondary structure motion (6N8X) |
|---|---|---|
| **a** | **c** | **e** |
| ligand RMSD: 1.283 △pocket RMSD: -0.669, △clash: -0.445 | ligand RMSD: 1.718 △pocket RMSD: -1.140, △clash: -0.165 | ligand RMSD: 0.558 △pocket RMSD: -2.297, △clash: -0.660 |
| Gate opening (6UWV) | Secondary structure change (6ROT) | Interdomain motion (6S9X) |
| **b** | **d** | **f** |
| ligand RMSD: 0.866 △pocket RMSD: -0.465, △clash: -0.406 | ligand RMSD: 0.466 △pocket RMSD: -2.327, △clash: -0.356 | ligand RMSD: 2.835 △pocket RMSD: -5.245, △clash: -0.192 |

ps    ns    μs    ms

**Fig. 4 | DynamicBind effectively captures protein dynamics across diverse time scales.** Proteins undergo conformational changes that can occur across a range of time scales upon binding with small-molecule ligands. A negative Δpocket RMSD indicates the predicted structure has a lower RMSD with the ground truth relative to the AlphaFold structure. A negative Δclash implies that the predicted ligand has lower clash score with the predicted structure compared to the transplanted ligand with the AlphaFold structure. **a** The side chain of Arginine rotates, mitigating clashes with the ligand. **b** The Tyrosine in the AlphaFold structure that was obstructing the binding pocket, shifts away in the predicted structure. **c** The loop region of the AlphaFold structure intersects with the ligand, and it is re-positioned in the predicted structure. **d** Alpha helices near the binding site transform into loops, aligning with the crystal structure. **e** The alpha helix of Hsp90 experiences a considerable relocation. **f** Two domains coalesce, thereby forming the binding pocket. Source data are provided as a Source Data file.

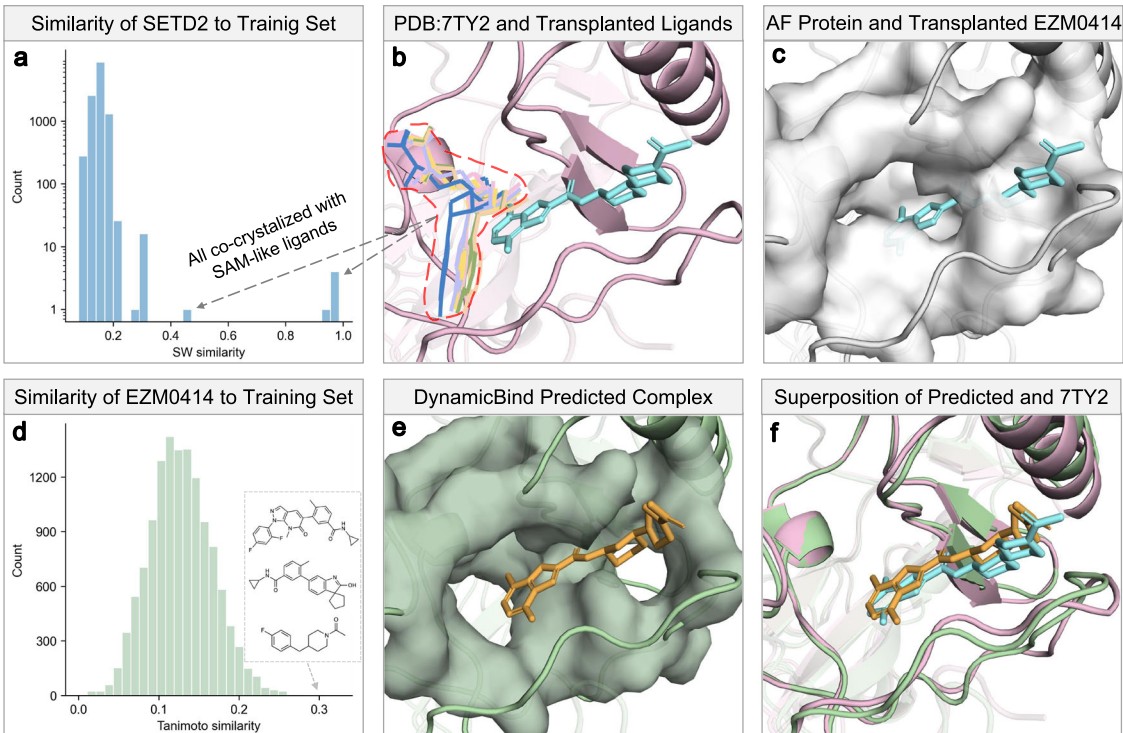

**Fig. 5 | DynamicBind reveals cryptic pocket for ligand EZM0414. a** Only six PDBs in the training set have a protein Smith–Waterman similarity greater than 0.4 with the SETD2 protein, and all are co-crystallized with SAM-like ligands, also shown in lines in (**b**). The ligand of PDB 7TY2, EZM0414, is displayed in cyan sticks, with the protein shown in pink. **c** The binding pocket for EZM0414 is absent in the AlphaFold structure, depicted in white. **d** This panel shows the Tanimoto similarity of ligands in the training set compared to EZM0414, and the top three most similar ligands are drawn out. **e** The protein–ligand complex structure as predicted by DynamicBind, with the protein represented in green and the ligand in orange. **f** The superposition of the complex as predicted by DynamicBind and the corresponding crystal structure. Source data are provided as a Source Data file.

been crystallized with a compound bound at the site targeted by EZM0414 (depicted in cyan sticks). Consequently, our model had not been trained on any structures with a compound bound to this newly identified site. In Fig. 5c, the AlphaFold structure and its surface are shown in white. The cryptic site appears blocked, causing substantial clashes with the transplanted EZM0414. Figure 5d confirms EZM0414 as an unseen ligand, with even the most similar Tanimoto ligands deviating substantially from EZM0414. Figure 5e displays the protein–ligand complex structure predicted by our model, taking the AlphaFold-predicted structure of SETD2 and the SMILES representation of EZM0414 as inputs. Figure 5f overlays our prediction with the crystal structure of the SETD2-EZM0414 complex (PDB 7TY2). The resultant ligand RMSD is 1.4 Å, and the pocket RMSD is 2.16 Å. Furthermore, we have included in the Supplementary Information several cases from the Cryptosite dataset[46] that have low sequence similarity to our training set.

### DynamicBind achieves better screening performance in an antibiotics benchmark

In target-based drug discovery, both screening of potential drug candidates and reverse screening, where protein targets are identified for specific compounds, are crucial. These processes require accurate prediction of binding affinities, the measure of the interaction strength between a protein and a compound, at a proteome level. Therefore, we have added an affinity prediction module to our model, trained using experimentally measured binding affinity data from the PDBbind dataset. To assess DynamicBind in a real-world virtual screening scenario, we used a recently published antibiotic experimental benchmark[25]. This dataset includes a panel of 2616 protein-compound pairs, none of which were encountered during our training phase. It features 12 proteins from the essential proteome of Escherichia coli

paired with 218 active antibacterial compounds. Figure 6a shows that DynamicBind surpasses both common docking methods like VINA and DOCK6.9 and the best machine learning-based re-scoring methods, achieving the mean average area under the receiver operating characteristic curve (auROC) of 0.68. Baseline numbers are directly sourced from the benchmark paper[25]. This performance improvement is due to DynamicBind's dynamic docking capability, which refines the AlphaFold structure towards a more native-like state, leading to a more precise binding affinity estimation. As depicted in Fig. 6b, the predicted structures of protein murD conform more closely around the ligand, forming more interactions that were not possible with the initial AlphaFold structure. This evaluation on the antibiotics benchmark agrees with our benchmarks on PDBbind test sets for binding affinity predictions (Supplementary Table 1), where DynamicBind consistently outperforms traditional docking methods and deep learning-based rigid docking methods. These results indicate that DynamicBind, with its binding affinity prediction capability, exhibits significant potential for proteome-level virtual screening applications.

## Discussion

DynamicBind unifies two conventionally separated steps, protein conformation generation, and ligand pose prediction, into a single framework. As an end-to-end deep learning method, it is orders of magnitude faster than traditional MD simulations in sampling extensive protein conformational changes. Unlike traditional docking methods that demand predefined binding pockets, DynamicBind has the capability to perform global docking, a feature that becomes essential when the binding pocket has yet to be identified. These advantages empower DynamicBind for the virtual screening of compounds that bind to cryptic pockets. Such compounds are likely to bind exclusively to the target protein, thereby potentially minimizing

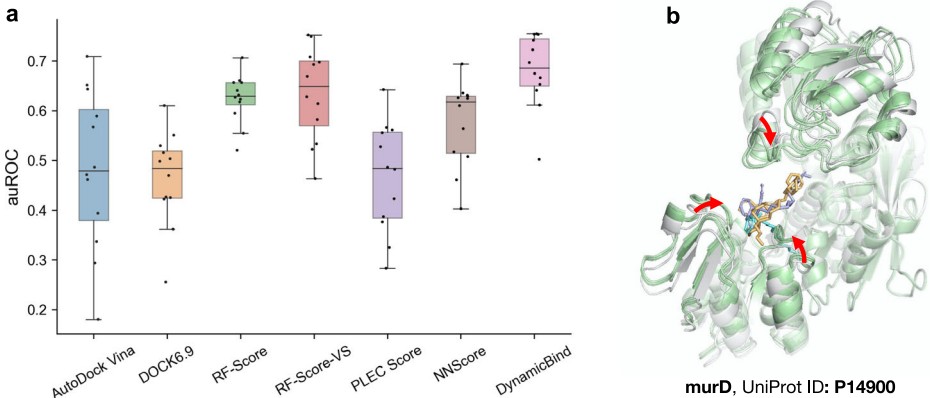

**murD**, UniProt ID: **P14900**

**Fig. 6 | DynamicBind achieves better screening performance in an antibiotics benchmark. a** Comparative evaluation of the virtual screening performance on the antibiotics benchmark by different methods, measured in terms of auROC (area under the ROC curve). The benchmark encompasses $n = 12$ distinct protein systems. Each box plot shows the median (central line), upper and lower quartiles (box edges), whiskers extending up to 1.5 times the interquartile range, and individual data points (dots). We faithfully incorporated all six baseline numbers as presented in the benchmark paper[25]. **b** The AlphaFold-predicted protein structure is shown in white, while the protein structures generated by DynamicBind for three active compounds are shown in green. Red arrows indicate the regions where the protein moves closer to the ligand, forming additional interactions. Source data are provided as a Source Data file.

side effects. In addition, DynamicBind can predict whether a new drug candidate may bind to an unintended protein target or can aid in identifying the binding target when an active compound is discovered via phenotype screening.

DynamicBind, while demonstrating state-of-the-art performance in our benchmarks, still presents opportunities for improvement, especially in enhancing its ability to generalize to proteins with low sequence homology compared to those in the training set[47,48]. As a data-driven model, it significantly benefits from rapid advancements in Cryo-EM methods[49–51]. These technological progressions will broaden the diversity and comprehensiveness of our training data, providing more varied conformations of protein–ligand complexes at a faster rate. There is also potential to improve DynamicBind by utilizing a large amount of non-structural binding affinity data, which are currently more abundant than crystallized structures. By adopting a self-distillation approach analogous to AlphaFold[5], we could augment our training set by integrating high-confidence predictions of the complex structures of protein–ligand pairs that previously only had affinity data available.

In summary, DynamicBind presents a "dynamic docking" approach for investigating protein–ligand interactions, setting it apart from traditional docking methods that treat proteins as static and molecular dynamics (MD) simulations that are computationally demanding. Its capacity for large-scale protein dynamics carries particularly significant implications for the discovery of drug molecules, especially those targeting cryptic pockets. In addition, the ligand-specific protein conformations generated by DynamicBind may offer valuable insights into the influence of ligands on proteins, potentially clarifying structure-function relationships and augmenting our mechanistic understanding.

## Methods
### Overview
Our model is an E(3)-equivariant, diffusion-based, graph neural network utilizing a coarse-grained representation.

An E(3)-equivariant model transforms the output, $y$, according to the trans-rotation and parity operations applied to the input $x$ in 3D space[52]. Research has demonstrated that equivariant models can be trained with 1000 times less data while yielding superior results on the structures of bulk water[53]. Despite substantial advancements in cryo-electron microscopy and crystallography, the existing protein–ligand complex database remains relatively limited, only extending to tens of

thousands in size. Consequently, an efficient model is required, capable of discerning the most relevant information and avoiding superficial information that does not hold true upon relocating or rotating the entire structure. The traditional approach to fulfilling the SO(3) symmetry involves exclusively using or predicting invariant quantities, such as the contact map. However, a contact or distance map does not always correlate with physically feasible configurations. For instance, a residue may be predicted to be in contact with two vastly distant atoms. In addition, a contact map may overlook chirality, a significant aspect in drug discovery[54].

As a diffusion-based model, DynamicBind is trained through a process that incrementally distorts the native conformation at various degrees, enabling the model to learn how to restore the correct conformation. Distorting the original configuration commonly involves adding trans-locational Gaussian noise to the atoms. With bond distance constraints imposed by chemical bonds and excluded volume effects enforced by Van der Waals forces, restoring from such distortions is straightforward when the distortion is relatively small. However, we observed that merely adding Gaussian noise is insufficient to train a model that can predict the transformation from one biologically meaningful configuration to another. To address this, we introduced a morph-like transformation that interpolates between the crystal protein structure and the structure predicted by AlphaFold, thereby reducing the transition barriers between meta-stable configurations, such as the AlphaFold-predicted conformation, and the ligand-bounded holo configuration. Unlike other generative models that train a score function, $s_\theta(\mathbf{x},t) \approx \nabla \log p_t(\mathbf{x})$, our diffusion architectures aim to map perturbed structures directly back to the original conformations, akin to the consistency model[29,55]. The outputs of the model are denoted as $f_\theta(\mathbf{x}_t,t) = -\boldsymbol{\phi}(\mathbf{x}_t,t)$, where $\boldsymbol{\phi}(\mathbf{x}_t,t)$ represents the added morph-like transformation to the native conformation.

Traditional methods use an all-atom representation, modeling the coordinates of every atom explicitly. However, atoms do not move independently due to their connections via chemical bonds, and local geometry is highly constrained—for example, a benzene ring is generally flat. To reduce the number of degrees of freedom of these nonphysical configurations, we adopted a coarse-grained representation for both the protein and the ligand. In our model, each protein residue is represented by a node with two vectors—coordinates and directions, and side-chain dihedral angles. More details are provided in "Featurization". For the ligand, every heavy atom is represented by a node, and these nodes transform in an extrinsic-to-intrinsic manner,

wherein changes in torsional angles are converted into changes in Cartesian coordinates[33]. Additional details can be found in "Transformation of the ligand conformation". Notably, despite being a coarse-grained representation, the coordinates of all non-hydrogen atoms can still be mapped in a one-to-one manner.

The input to our model is the current conformation of the protein and the ligand. The outputs include the predicted updates to $k^l$ scalar torsion angles and two translation–rotation vectors for each ligand, along with updates to $k_i^p$ scalar dihedral angles of the side-chain and two translation–rotation vectors of the backbone for each protein residue. Further details can be found in "Transformation of the protein conformation". In addition, the model produces two scalar outputs: one to estimate the degree of the native conformation as assessed by cLDDT (contact-LDDT), and another to predict the binding affinity between the protein and ligand.

## Featurization

The ligand in our model is the attributed graph $\mathcal{G}^l = (\mathcal{V}^l, \mathcal{E}^l)$, in which each node $v_i^l \in \mathcal{V}^l$ represents a heavy atom and the aromatic, single, double, or triple bonds as the edges. The node features of the ligand graph include atomic number, chirality, degree, and formal charge. In addition to bond type, edge length embedding is also used as scalar edge features.

The protein graph is denoted as $\mathcal{G}^p = (\mathcal{V}^p, \mathcal{E}^p)$, where each node $v_i^p \in \mathcal{V}^p$ corresponds to a residue at the $C_\alpha$ position. The node features in the protein graph include amino acid type, language model embedding from **esm**[7], and side-chain dihedral angles, which are represented as $(7 \times 2)$-dimensional zero-padded scalar features (five rotatable chi angles [chi1, chi2, ..., chi5] and two symmetric chi angles [altchi1, altchi2] for each amino acid, and these angles are transformed into sine and cosine values). To ensure the uniqueness of the side-chain angles for a given structure, we consistently handle it as [max( chi1,altchi1), min(chi1,altchi1), max(chi2,altchi2), min(chi2, altchi2 ), chi3, chi4, chi5]. In addition, the backbone orientation is represented as two unit vector features, which are $\frac{\mathbf{x}_N - \mathbf{x}_{C_\alpha}}{\|\mathbf{x}_N - \mathbf{x}_{C_\alpha}\|}$ and $\frac{\mathbf{x}_C - \mathbf{x}_{C_\alpha}}{\|\mathbf{x}_C - \mathbf{x}_{C_\alpha}\|}$. For edges, length embedding is used as scalar features. Our featurization of the amino acid enable the model to infer the positions of all heavy atoms.

## Model architecture

DynamicBind is a graph neural network that uses both equivariant and invariant features. It propagates information using tensor products of irreducible representations (irreps) as per the definitions in the e3nn library[52].

The input scalar features of nodes and edges are concatenated with sinusoidal embeddings[56] of diffusion time and then encoded by different multilayer perceptrons (MLPs). For the protein node, the two unit vector features of amino acids are combined with the new scalar representations to form the initial features for interaction layers. Similar to DiffDock[19], in each step of the graph propagation process, the ligand and protein graphs undergo one intra-interaction and one inter-interaction. In the ligand's intra-interaction, the representation of each ligand atom is updated by other ligand atoms within a distance of 5 Å. For the protein, each amino acid is updated by other amino acids within a distance of 15 Å. To reduce the training runtime and memory usage of the model, a maximum of 24 neighbors is allowed for each residue. The edges for inter-interaction are determined based on whether an amino acid is within a distance of $(3\sigma_{tr} + 12)$ Å from any ligand atom, where $\sigma_{tr}$ is the current standard deviation of the diffusion translational noise. This dynamic cutoff is designed to ensure interconnections exist even when the

ligand is far from the receptor when $\sigma_{tr}$ is large. After the connected graph is determined, the messages of the node is updated by the TensorProductLayer. Specifically, for each node a belonging to category $c_a$:

$$\mathbf{h}_a \leftarrow \mathbf{h}_a \underset{c \in \{\ell, r\}}{\oplus} \mathrm{BN}^{(c_a, c)} \left( \frac{1}{|\mathcal{N}_a^{(c)}|} \sum_{b \in \mathcal{N}_a^{(c)}} Y(\mathbf{r}_{ab}) \otimes_{\psi_{ab}} \mathbf{h}_b \right) \quad (1)$$

$$\text{with } \psi_{ab} = \Psi^{(c_a, c)} \left( e_{ab}, \mathbf{h}_a^0, \mathbf{h}_b^0 \right)$$

Here, $\mathbf{h}_a$ represents the features of a node, and $\mathbf{h}_a^0$ denotes its scalar features. $\mathcal{N}_a^{(c)}$ refers to the neighbors of node a of category c (either ligand, or protein). The spherical harmonics are denoted as $Y$, and BN represents the (equivariant) batch normalization. The module $\Psi$ is a MLP which contains learnable weights for the tensor product, which are computed based on the edge embeddings, $e_{ab}$, and scalar features, $\mathbf{h}_a^0, \mathbf{h}_b^0$.

After the final interaction layer, the node representations are used to produce the outputs. For generating the cLDDT, binding affinity, ligand's translation and rotation predictions, a convolution of each ligand atom with the geometric center of the ligand is employed:

$$\mathbf{v} = \frac{1}{|\mathcal{V}^\ell|} \sum_{a \in \mathcal{V}^\ell} Y(\mathbf{r}_{oa}) \otimes_{\psi_{oa}} \mathbf{h}_a \quad (2)$$

$$\text{with } \psi_{oa} = \Psi \left( e_{oa}, \mathbf{h}_a^0 \right)$$

where $e_{oa}$ is the edge embedding between the geometric center of the ligand and a ligand node a. The output $\mathbf{v}$ consists of 144 even scalars, 2 odd parity vectors and 2 even vectors. The scalars are used for predicting the cLDDT (D) and negative logarithm of the binding affinity (A) as measured in the unit of concentration.

$$D = \mathrm{MLP} \left( \mathbf{v}_{scalar}[: 72] \right) \quad (3)$$

$$A = \mathrm{clamp} \left( \frac{\mathrm{MLP}(\mathbf{v}_{scalar}[72:144])}{D + \mathrm{eps}}, \min = 0, \max = 15 \right) \quad (4)$$

The odd vectors are used to predict ligand translation, while the even vectors are used to predict ligand rotation:

$$\mathbf{tr}^l = \frac{\bar{\mathbf{v}}_{vector}^{odd}}{\| \bar{\mathbf{v}}_{vector}^{odd} \| + \mathrm{eps}} \times \mathrm{MLP} \left( \| \bar{\mathbf{v}}_{vector}^{odd} \|, \mathbf{s}_t \right) \quad (5)$$

$$\mathbf{rot}^l = \frac{\bar{\mathbf{v}}_{vector}^{even}}{\| \bar{\mathbf{v}}_{vector}^{even} \| + \mathrm{eps}} \times \mathrm{MLP} \left( \| \bar{\mathbf{v}}_{vector}^{even} \|, \mathbf{s}_t \right) \quad (6)$$

$$\text{with } \bar{\mathbf{v}}_{vector} = \frac{\mathbf{v}_{vector}[0] + \mathbf{v}_{vector}[1]}{2}$$

Here, $\mathbf{s}_t$ is the sinusoidal embeddings of the diffusion time, eps $= 10^{-12}$ is added for numerical stability. Following Jing et al.[33], our model predicts a scalar torsion update for each rotatable bond of ligand. For bond b, the torsion update $T_b^l$ is generated by a convolution of every atom on a radius graph with the bond center o:

$$T_b^l = \mathrm{MLP} \left( \frac{1}{|\mathcal{N}_b|} \sum_{a \in \mathcal{N}_b} Y(\mathbf{r}_{oa}) \otimes Y^2(\mathbf{r}_b) \otimes_{\gamma_{oa}} \mathbf{h}_a \right) \quad (7)$$

$$\text{with } \gamma_{oa} = \Gamma \left( e_{oa}, \mathbf{h}_a^0, \mathbf{h}_{b_0}^0 + \mathbf{h}_{b_1}^0 \right)$$

To predict the conformation changes of protein, we require updates of the side-chain chis, translation, and rotation for each protein node. These operations are generated from the final interaction

representations $\mathbf{h}_i$ of each amino acid:

$$T_i^p = \text{MLP}\left(\mathbf{h}_{i,scalar}^{odd}, \mathbf{h}_{i,scalar}^{even}\right) \tag{8}$$

$$\mathbf{tr}_i^p = \frac{\overline{\mathbf{h}}_{i,vector}^{odd}}{\parallel \overline{\mathbf{h}}_{i,vector}^{odd} \parallel + \text{eps}} \times \text{MLP}\left(\parallel \overline{\mathbf{h}}_{i,vector}^{odd} \parallel, \mathbf{s}_t\right) \tag{9}$$

$$\mathbf{rot}_i^p = \frac{\overline{\mathbf{h}}_{i,vector}^{even}}{\parallel \overline{\mathbf{h}}_{i,vector}^{even} \parallel + \text{eps}} \times \text{MLP}\left(\parallel \overline{\mathbf{h}}_{i,vector}^{even} \parallel, \mathbf{s}_t\right)$$
$$\text{with } \overline{\mathbf{h}}_{i,vector} = \frac{1}{|\mathcal{N}_i|}\sum_{j\in\mathcal{N}_i}\mathbf{h}_{i,vector}^j \tag{10}$$

Here, $T_i^p$ is a five-dimensional scalar output representing torsion updates for [chi1, chi2, ..., chi5].

## Transformation of the ligand conformation

To update the ligand conformation, we employ a unified global translation $\mathbf{tr}^l \in \mathbb{R}^3$ and rotation $R^l \in \mathbb{R}^{3\times3}$. All atoms of the ligand will be simultaneously translated and rotated around the geometric center of the ligand, which is calculated as $\overline{\mathbf{x}}^l = \frac{1}{n}\sum \mathbf{x}_i^l$, where $n$ is the total number of heavy atoms of the ligand and $\mathbf{x}_i^l$ denotes the position vector of atom $i$. Specifically, the transformed position vector $\mathbf{x}^l$ is obtained as $\mathbf{x}^l = R^l(\mathbf{x}^l - \overline{\mathbf{x}}^l) + \overline{\mathbf{x}}^l + \mathbf{tr}^l$.

In addition to translation and rotation, torsion angles are also crucial factors in determining the ligand conformation. However, modifying torsion angles can perturb the position of the center of mass of the ligand. To address this issue, Corso et al.[19] demonstrated that performing an RMSD alignment after updating the torsion angles can ensure that the effect of the torsion updates is orthogonal to the roto-translation updates, and thus decouple the consequences of torsional updates and roto-translation updates. Overall, the updated ligand pose is obtained as $\mathbf{x}^l = \text{RMSDAlign}((T_0^l \circ \cdots T_k^l)(\mathbf{x}^l), R^l(\mathbf{x}^l - \overline{\mathbf{x}}^l) + \overline{\mathbf{x}}^l + \mathbf{tr}^l)$, where $T_k^l$ is the torsion rotation.

## Transformation of the protein conformation

Following AlphaFold[5], we use C$\alpha$ as the residue node to perform global translation and rotation. Additionally, the model predicts the updates of side-chain torsion angles. For 180°-rotation-symmetric side-chain parts, considering symmetry is unnecessary in the inference stage, but we introduce symmetry side-chain torsion features during training to correctly compute the loss function. Since the position of the C$\alpha$ is independent of the side-chain torsion angles, rotating the side chain does not affect the residue-level translation and rotation. Thus, we can perform roto-translations and torsion rotations in any order. Finally, the updated conformation of each protein residue is represented as $\mathbf{x}_i^p = (T_{i,0}^p \circ \cdots T_{i,k}^p)(R_i^p(\mathbf{x}_i^p - \mathbf{x}_{i,c_\alpha}^p) + \mathbf{x}_{i,c_\alpha}^p + \mathbf{tr}_i^p)$, where $T_{i,k}^p$ is the side-chain torsion rotation of $i$th residue.

## Training and inference

During the training process, the input are the protein structure in decoy conformation constructed by adding morph-like transformation to the native conformation and the ligand structure in conformation with Gaussian noise added. The expected output are the denoising operations. The input protein structure at time t is defined as $\mathbf{x}_t^p = \phi(\mathbf{x}^{holo}, t)$. Specifically, for the $i$th amino acid, the Kabsch algorithm[57] is used to calculate the translation $\mathbf{tr}_i^*$ and rotation $\mathbf{rot}_i^*$ around C$\alpha$ that aligns the backbone atoms $N - C\alpha - C$ of the holo-structure to the apo structure:

$$\mathbf{tr}_i^*, \mathbf{rot}_i^* = \text{Kabsch}\left(\mathbf{x}_{i,(N,C_\alpha,C)}^{holo} - \mathbf{x}_{i,C_\alpha}^{holo}, \mathbf{x}_{i,(N,C_\alpha,C)}^{apo} - \mathbf{x}_{i,C_\alpha}^{holo}\right) \tag{11}$$

Considering the differences in torsion angles, we can draw the conformation changes of $i$th residue:

$$\mathbf{x}_i^{apo} = \phi(\mathbf{x}_i^{holo}) = (T_{i,0}^* \circ \cdots T_{i,k}^*)\left(R_i^*\left(\mathbf{x}_i^{holo} - \mathbf{x}_{i,C_\alpha}^{holo}\right) + \mathbf{x}_{i,C_\alpha}^{holo} + \mathbf{tr}_i^*\right) \tag{12}$$

Here, $T_{i,k}^* = T_{i,k}^{apo} - T_{i,k}^{holo}$ are in radian and the $R_i^*$ is the rotation matrix of $\mathbf{rot}_i^*$. At any given moment, we aim to perturb the protein structure using a factor, denoted as $u(t)$, such that the perturbed data is an intermediate state between the holo-structure and apo structure:

$$\phi\left(\mathbf{x}^{holo}, t\right) = \left(\Delta T_{i,0}^p \circ \cdots \Delta T_{i,k}^p\right)\left(\Delta R_i^p\left(\mathbf{x}_i^{holo} - \mathbf{x}_{i,C_\alpha}^{holo}\right) + \mathbf{x}_{i,C_\alpha}^{holo} + \Delta\mathbf{tr}_i^p\right)$$
$$\text{with} \quad \Delta\mathbf{tr}_i^p = u(t)\mathbf{tr}_i^*$$
$$\Delta R_i^p = \text{Rotation matrix of } u(t)\mathbf{rot}_i^*$$
$$\Delta T_{i,k}^p = u(t)T_{i,k}^* + \mathcal{N}(0,0.3)$$
$$u(t) = \text{clamp}\left(\tau_{min}^p + (\tau_{max}^p - \tau_{min}^p)\times(5t)^{0.3}, \min=0, \max=1\right) \tag{13}$$

where $\tau_{min}^p$ and $\tau_{max}^p$ represent the parameters of the diffusion noise.

To overcome the distribution shift between training and inference that arises from the use of RDKit-generated conformations as starting points in the inference process, we replace the training objective with the conformation $\mathbf{x}_0^l$ that matched to the ground truth pose $\mathbf{x}^{gt}$[19,33]. At time $t$, the input ligand pose is a random perturbed conformation:

$$\mathbf{x}_t^l = (\Delta T_0^l \circ \cdots \Delta T_k^l)(\Delta R^l(\mathbf{x}_0^l - \overline{\mathbf{x}}_0^l) + \overline{\mathbf{x}}_0^l + \Delta\mathbf{tr}^l)$$
$$\text{with} \quad \Delta\mathbf{tr}^l = \left(\mathcal{N}\left(0,\sigma_{tr}^l\right), \mathcal{N}\left(0,\sigma_{tr}^l\right), \mathcal{N}\left(0,\sigma_{tr}^l\right)\right)$$
$$\Delta R^l = \text{Rotation matrix of sampling from } p(\omega)\hat{\omega}$$
$$\Delta T_k^l = \mathcal{N}\left(0,\sigma_{tor}^l\right)$$
$$p(\omega) = \frac{1-cos(\omega)}{\pi}\sum_{l=0}^{\infty}(2l+1)\exp\left(-l(l+1)(\sigma_{rot}^l)^2\right)\frac{\sin((l+1/2)\omega)}{\sin(\omega/2)} \tag{14}$$

Here, $\overline{\mathbf{x}}_0^l$ is the geometric center of $\mathbf{x}_0^l$, $p(\omega)$ is the isotropic Gaussian distribution on SO(3) and the $\hat{\boldsymbol{\omega}}$ is a unit vector generated by random sampling.

The network is trained with eight losses. The total loss can be defined as follows

$$\mathcal{L} = \tfrac{1}{3}\mathcal{L}_{\mathbf{tr}}^l + \tfrac{1}{3}\mathcal{L}_{\mathbf{rot}}^l + \tfrac{1}{3}\mathcal{L}_T^l + \tfrac{1}{3}\mathcal{L}_{\mathbf{tr}}^p + \tfrac{1}{3}\mathcal{L}_{\mathbf{rot}}^p + \tfrac{1}{3}\mathcal{L}_T^p + 0.01\mathcal{L}_A + 0.99\mathcal{L}_D \tag{15}$$

where $\mathcal{L}_{\mathbf{tr}}^l, \mathcal{L}_{\mathbf{rot}}^l, \mathcal{L}_T^l$ are the losses for the translation, rotation, and torsion of the ligand, respectively. The $\mathcal{L}_{\mathbf{tr}}^p, \mathcal{L}_{\mathbf{rot}}^p$, and $\mathcal{L}_T^p$ are the losses for the protein residues. The $\mathcal{L}_A$ is binding affinity loss and the $\mathcal{L}_D$ is contact-LDDT loss. The distance difference for computing the ground-truth cLDDT is $d = |d(\mathbf{x}_0^l, \mathbf{x}^{holo}) - d(\mathbf{x}_t^l, \mathbf{x}_t^p)|$ (more details of the cLDDT score calculation can be found in "Evaluation metrics").

Since a rotation vector $\mathbf{u}$ represents the same rotation as another $\mathbf{v}$ if $\mathbf{u}$ and $\mathbf{v}$ have opposite orientation and $\|\mathbf{u}\| + \|\mathbf{v}\| = 2\pi$. So we take the minimum of the forward and opposite orientation losses when computing the rotation loss. The torsion angle losses are computed using the cosine of the angle difference between the predicted value and the added torsion angle noise. The full training procedures can be see in Supplementary Algorithm 1.

During the inference process, we use the ligand structure with conformations generated by RDKit and the protein structure prediction by AlphaFold as the initial complex conformation. The complex structure is updated with 20 steps. To prevent the final conformation trapped in local minimum, in each step, a small random noise is added to the denoised ligand pose. For each pair, we perform 40 samplings and rank the binding conformations based on the predicted cLDDTs.

We also noticed that the weighted averaged of the predicted binding affinity is a more accurate estimator of the experimentally measured affinity (Supplementary Table 1). The predicted cLDDT values is used as the weights. The complete inference procedures can be found in Supplementary Algorithm 2.

DynamicBind has 63.67 million parameters and was trained for 5 days on eight Nvidia A100 80GB GPUs.

## Evaluation metrics

To assess the interaction between the protein and the ligand within the predicted complex structure, we determine the extent of intermolecular native contact formation. We adopt a definition similar to that of the Local Distance Difference Test (LDDT) score, previously employed for quantifying the nativeness of predicted protein structures[58]. The Contact-LDDT (cLDDT) score is computed by considering the distances less than 15 Å among all pairs of ligand atoms and protein atoms. The distance difference is determined between the ground truth and the predicted complex structure, while accounting for symmetry. The final cLDDT score is derived from the mean fraction of conserved distances across four tolerance thresholds: 0.5, 1, 2, and 4 Å.

In order to evaluate the deviation of the predicted protein structure from the native protein structure surrounding the binding pocket, we compute the pocket Root Mean Square Deviation (pocket RMSD). This is performed using protein atoms located within 5 Å of the reference ligand atoms. Initially, the predicted protein structure is aligned with the crystal protein structure. Subsequently, the RMSD between the predicted pocket atoms and the crystal pocket atoms is determined.

Similiar to AlphaFill[14], the clash score is the root mean square (RMS) of the van der Waals overlaps[59] across all distances between the ligand atoms and the protein atoms, which are less than 4 Å. It is computed as follows:

$$\text{clash score} = \sqrt{\frac{\sum_{i=0}^{N} \text{VdW overlap}_i^2}{N}} \qquad (16)$$

where $N$ is the number of distances considered.

## Dataset construction

Our training and test dataset was built upon the PDBbind2020[34] database, which includes a curated collection of 19,443 crystal structures of protein–ligand complexes, each paired with an experimentally measured binding affinity. We employed the same time split as previous works[19,35,36], using structures deposited before 2019 for training and validation, while those deposited in 2019 were reserved for testing. Each protein was aligned with the AlphaFold-predicted structure that corresponds to the same protein sequence. The aligned AlphaFold structures and the crystal structures are used to generate training samples of the protein part through morph-like interpolation. The Major Drug Targets (MDT) test set was constructed using the following criteria: PDBs deposited in 2020 or later; proteins belonging to one of the four major drug target groups - kinases, GPCRs, nuclear receptors, and ion channels; the AlphaFold-predicted protein structures have pocket RMSD above the 2 Å (or pocket LDDT below 0.8) with the crystal structure; ligands are drug-like small molecules with molecular weights between 200 and 650 Dalton; at most 10 PDBs from a single study are included. These criteria ensure that the test set is challenging, with the initial input protein differs from the native conformation, and is representative, covering a wide range of protein targets. In addition, it prevents a few proteins dominating the entire test set, as certain studies deposited significantly more PDBs, structures of the same protein co-crystallized with slightly different ligands, than other studies.

## Baselines

We performed docking on both PDBbind test set (303 ligand-receptor pairs) and Major drug targets (MDT) test set (599 ligand-receptor pairs) using different docking methods listed below. The docking ligands were extracted from the co-crystalized structures without changing their atomic coordinates and the docking receptor structures were predicted by AlphaFold. We use a symmetry-aware method, specifically the symmrmsd function from the spyrmsd package[60] for all RMSD computations.

### Autodock VINA rigid

In Autodock Vina[17], ligands were converted from SDF format to PDBQT format by Meeko 2.0.0. Protein preparation was performed by using the "prepare_receptor" command in ADFR Suite 1.0. The docking box was defined using an automatic box around the native ligand with the default buffer of 4 Å on all six sides. And the box center was the center of mass of the native ligand. Because the boron atom is not a valid AutoDock atom type, ligands with this atom cannot be docked. Therefore, only 301 ligand-receptor pairs in PDBbind dataset and 597 ligand-receptor pairs in MDT dataset had docking output in VINA rigid docking.

### Autodock VINA flex

Comparing to VINA rigid docking, there is an additional flexible receptor preparation step in VINA flexible docking. It was performed by a python script called "prepare_flexreceptor.py", which is available at https://github.com/ccsb-scripps/AutoDock-Vina/tree/develop/example/autodock_scripts. Through this step, the protein PDBQT format file was divided into two PDBQT format files, one for the rigid part and one for the flexible side chains. For Vina Flex mode, flexible side chains must be predetermined. We identified all residues with side-chain atoms within 5 Å of the ligand atoms as flexible. In this mode, the protein backbone remains rigid. Ligand preparation and grid box setting were consistent with VINA rigid docking.

### GNINA rigid

The ligand input files for GNINA[61] are in PDBQT format, created using OpenBabel after adding hydrogens with RDKit Protein input files were PDB format files. The grid box setting was consistent with VINA rigid docking. For the PDBbind dataset, all of the ligand-receptor pairs had docking output. For MDT dataset, 1 pair had no output because the ligand in the original PDB file (PDB ID: 8HMU) was not completely resolved, which had missing atoms.

### GLIDE

GLIDE[16] is a rigid protein docking module in Schrödinger software. Ligands were prepared by using the LigPrep module. Protein preparation was performed by using the Protein Preparation Wizard module. Grid files were generated by the Receptor Grid Generation module with a 10 Å inner box and an automatic outer box around the ligand with the default buffer of 4 Å on all six sides centered on the center of mass of the ligand. Then, the SP precision docking was performed. Some of the ligands in PDBbind dataset are polypeptides, which cannot be processed by LigPrep module. In addition, ligands with severe clashes with pocket atoms had no output pose during docking. Therefore, 266 ligand-receptor pairs in PDBbind dataset and 472 ligand-receptor pairs in MDT dataset had docking output in GLIDE rigid docking.

### Induced fit docking

Induced fit docking (IFD) module[62] in Schrödinger software provides a protein-flexible docking function for the user. Different from VINA and GNINA, not only residue side chains but also residue backbones can move slightly. Ligand preparation and protein preparation were the same as GLIDE rigid docking. The search space was defined by default parameters, a 10 Å inner box and an outer box with auto size (similar in size to ligand) centered on the center of mass of the native ligand.

Amino acid residues within 5 Å of the ligand atoms were defined as flexible residues. The docking process was performed under the standard protocol, which generates up to 20 poses. In total, 284 ligand-receptor pairs in PDBbind dataset and 580 in MDT dataset were docked successfully by using the IFD module. Induced fit docking can give output poses successfully for more ligand-receptor pairs in PDBbind dataset than GLIDE rigid docking, indicating that this docking method can extend the pocket by moving pocket residues.

## Reporting summary

Further information on research design is available in the Nature Portfolio Reporting Summary linked to this article.

## Data availability

Raw data were sourced from the public dataset PDBbind2020, available at http://www.pdbbind.org.cn/index.php. The data generated in this study and processed training data have been publicly deposited to Zenodo under https://doi.org/10.5281/zenodo.10429051. Source data are provided with this paper.

## Code availability

Demo, instructions, and codes for DynamicBind are available at https://github.com/luwei0917/DynamicBind. The version used for this publication is available in ref. 63. In addition, a web server is available at https://m1.galixir.com/#/home/demo/dynamicDocking.

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

## Acknowledgements

We are grateful to Meihui Song and Jiahui Tang for their visualization and inspiration. We thank Da Wei and Ziwei Huang for their assistance in web server setup and IT support. We thank James J. Collins for his valuable feedback and insightful discussions. S.Z. acknowledges funding from the Baidu scholarship. P.G.W. is supported by the Center for Theoretical Biological Physics sponsored by the NSF grant PHY-2019745 and the D.R. Bullard-Welch Chair at the Rice University Grant C-0016.

## Author contributions

S.Z. and W.L. conceived and supervised the project. W.L. and J.Z. contributed to the algorithm implementation. J.Z., W.L., S.Z., and W.H. contributed to the visualization and baseline implementation. W.L., S.Z., P.G.W., and J.Z. wrote the manuscript. All authors were involved in the discussions and proofreading.

## Competing interests

W.L., J.Z., W.H., Z.Z., X.J., Z.W., L.S., and C.L. work directly or indirectly for Galixir Technologies. S.Z. was a former employee of Galixir Technologies. The remaining authors declare no competing interests.
