## [Peer Review File · Nature Communications]

Reviewers' Comments:

Reviewer #1:

Remarks to the Author:

This paper describes an end-to-end equivariant deep learning model for flexible molecular docking. The model is trained to convert undocked ligands and alphafold predicted receptor structures into the bound complex. Overall, the paper is well written, there are several clever and novel ideas, and the results are exciting. I have only a few minor comments for the authors to consider.

Although overall I found the evaluation to be holistic (ranging from a systematic test set evaluation to more anecdotal, yet highly illustrative, evaluations), the paper would be stronger if more effort was made to measure generalization. What happens if the test set is filtered by similarity to training set (ideally this is assessed for different similarity cutoffs and shown with bootstrapped confidence intervals)? For most DL models, accuracy decreases significantly.

In Fig 2f it wasn't clear to me what the justification for having a flat line for Glide was (inconsistency in number of sample?). If only the top ranked pose is being plotted simply say that.

What is the rationale for the methods selected for Fig 6? Are they taken from the benchmark publication? If so, please state this explicitly as otherwise it seems like cherry picking since the methods from Fig 2 having gone missing. If not, please include the methods from Fig 2.

The morphing method reminds me of the diffusion inspired method of this paper:

<https://www.ncbi.nlm.nih.gov/pmc/articles/PMC10402188/>

If the authors agree, perhaps they could comment on similarities/differences (this paper does not do flexible docking).

Page 19: ground-turth

I am highly concerned about the statement in 4.4.3 that changing atom orders made it difficult to calculate RMSDs. All RMSD calculations should be done using symmetry aware methods like RDKit's CalcRMS function or OpenBabel's obrms tool, which would not have a problem with changing atom orders. Although I doubt this will change the results much (other than to make them better), there's always the possibility that the DL algorithm is learning preferred atom orders in a way that favors a non-symmetry corrected, atom order dependent RMSD calculation. Furthermore, as I anticipate this to be a highly influential paper, it is important to stick with best practices (and not correcting for symmetry is definitely not best practice when reporting a docking RMSD). Please redo your RMSD calculations using appropriate methods or clarify that you are already using them.

Reviewer #2:

Remarks to the Author:

The authors present the development and validation of a new diffusion-based neural network model for ligand docking to flexible proteins. The major innovation has been the utilization of a morphing function between known protein-ligand complex structures and AlphaFold structures instead of the usual use of Gaussian noise during training of diffusion models. The manuscript is quite interesting to read but I have a few major concerns that need to be addressed:

1. The authors split the dataset by time. This will result in a large overlap of homologous or even identical proteins in training and test set. In particular, regarding the study on the conformational change of kinases: It looks impressive that the model can predict DFG-in and DFG-out conformations. However, one has to recognize that this transition is conserved among the kinase family. Since there are many kinase structures in their training set, the model has learned the pattern of this transition (and not the underlying physics). Thus, it is not really surprising that the model is able to test both conformations during docking. This, however, is no prove of generalizability of the model.

In this context, the authors show an improvement over DiffDock (Figure 2). However, when we trained DiffDock on a dataset where training and test set proteins were split by maximum diversity

(to check generalizability) DoffDock drops to below 10% success rate (<2Å) demonstrating the existing bias in those type of diffusion models. As DynamicBind is based on the same architecture, it is very likely that also DynamicBind will drop significantly in performance when applying it to novel proteins. I would recommend a split which maximizes the difference of protein homology between training and test set.

2. In the Methods section the authors discuss the rigid and flexible version of Vina. In the Results section, however, I could only see one results column for Vina. Which version was used?

3. I do not think that the comparison with Vina, Glide and Smina is fair if they are run in the "rigid protein" mode. Glide, for example, offers a validated induced fit modus. Why did the authors not use this "flexible protein" modus for a fairer comparison with DynamicBind?

4. It is not clear if the classical docking methods were run with the docking box covering the full protein which typically is not successful. The more robust method for blind docking with those classical docking methods is to first predict all possible binding sites and then perform docking to all sites in sequence, combine the poses and rank them according to score. Otherwise, I again do not believe that the current comparison is fair.

5. Line 202: The authors state "In traditional docking methods, the sampling of protein conformations is decoupled from the docking step". This is not correct as methods such as Glide and Vina-flexible sample ligand and protein conformations (at least on some level) at the same time.

6. In chapter 2.5. the authors nicely show the potential of DynamicBind to identify ligands binding to cryptic sites. However, they only show a single result. How reproducible is this result for other systems? I would recommend that the authors select a few more PDB structures not homologous to any training set system from the Cryptosite website:

<https://modbase.compbio.ucsf.edu/cryptosite/> and repeat the predictions.

7. The authors provided a GitHub link. But no code is currently available at this location.

Dear Nature Communications Editors—

Thank you for the opportunity to revise our manuscript entitled, “DynamicBind: Predicting ligand-specific protein-ligand complex structure with a deep equivariant generative model” (NCOMMS-23-39499-T). Thanks also to the reviewers for their careful reading of our work and their thoughtful comments. The main text, Online Methods, and Supplementary Information have all been revised and extended accordingly. In particular, the Supplementary Information now includes additional new figures, analyses, and explanations that respond to the reviewers’ suggestions. We think that, as a result of answering the concerns of the reviewers, the manuscript has been substantially improved.

Our specific responses to the questions and concerns of each reviewer are given below. The original remarks of the reviewers are shown in black text and our responses are given in blue text. The corresponding changes in the manuscript and methods files have also been highlighted with blue text.

REVIEWER COMMENTS

Reviewer #1 (Remarks to the Author):

This paper describes an end-to-end equivariant deep learning model for flexible molecular docking. The model is trained to convert undocked ligands and alphafold predicted receptor structures into the bound complex. Overall, the paper is well written, there are several clever and novel ideas, and the results are exciting. I have only a few minor comments for the authors to consider.

We appreciate the reviewer’s positive appraisal of our work. We have addressed the reviewer’s specific comments below.

Although overall I found the evaluation to be holistic (ranging from a systematic test set evaluation to more anecdotal, yet highly illustrative, evaluations), the paper would be stronger if more effort was made to measure generalization. What happens if the test set is filtered by similarity to training set (ideally this is assessed for different similarity cutoffs and shown with bootstrapped confidence intervals)? For most DL models, accuracy decreases significantly.

We agree with the reviewer’s suggestion that stratifying the test set evaluations with varying similarity cutoffs would enhance the informativeness and robustness of our paper. We have added Section 5.3 of the Supplementary Information (SI) to include results now filtered at protein sequence similarity cutoffs of 30%, 60%, and 90% relative to the training set. We did observe a decrease in accuracy. We also found that other methods displayed a similar decrease in accuracy on the new test set. But notably, DynamicBind consistently outperformed the state-of-the-art deep learning method DiffDock at different similarity thresholds, highlighting its potential for enhanced generalization capability. We believe that with the availability of more experimental data, the model’s ability to generalize could improve significantly. We have updated the main text to reflect these changes. In Section 2.2, we now state, “Results for individual datasets, as well as those filtered at 30%, 60%, and 90% maximum sequence

similarity cutoffs, are detailed in Sections 5.2 and 5.3 of the Supplementary Information.” Additionally, in the discussion section, we have revised the text to include, “there is potential for further refinement, such as enhancing its generalization capabilities for proteins with low homology to the training set.”

In Fig 2f it wasn't clear to me what the justification for having a flat line for Glide was (inconsistency in number of sample?). If only the top ranked pose is being plotted simply say that.

Thank you for your question regarding Figure 2f. You're correct; the inconsistency in the number of samples outputted by Glide, likely due to the removal of unrealistic conformations through a filtering scheme, led to the decision to represent Glide with a flat line. This line indicates the success rate computed using the best sample from Glide. To clarify this in the main text, the text now reads “Due to the variability in the number of samples produced by Glide, often because of its filtering scheme eliminating unrealistic conformations, Glide's best performance is represented using a flat line. This line reflects the success rate determined by the most effective sample from Glide.”

What is the rationale for the methods selected for Fig 6? Are they taken from the benchmark publication? If so, please state this explicitly as otherwise it seems like cherry picking since the methods from Fig 2 having gone missing. If not, please include the methods from Fig 2.

Thank you for the observation. The selection of all six baseline methods in Figure 6 was indeed taken directly from the benchmark publication, as depicted in their Figure 5A (<https://www.embopress.org/doi/full/10.15252/msb.202211081>).

To clarify this in our paper, we have added the statement, “Baseline numbers are directly sourced from the benchmark paper [25].” in the main text. We also added “We faithfully incorporated all six baseline numbers as presented in the benchmark paper [25].” to the caption of Fig 6.

The morphing method reminds me of the diffusion inspired method of this paper: <https://www.ncbi.nlm.nih.gov/pmc/articles/PMC10402188/>

If the authors agree, perhaps they could comment on similarities/differences (this paper does not do flexible docking).

Thank you for pointing out this interesting paper. We've incorporated this reference into our main text discussion on diffusion-based methods. “Unlike the traditional protocol employed in diffusion-based model training, which generates decoys by perturbing the native state with Gaussian noise of varying magnitudes [28-32], our method employs a morph-like transformation to produce protein decoys.”

Page 19: ground-turth

The typo has been fixed. Great catch. Thanks!

I am highly concerned about the statement in 4.4.3 that changing atom orders made it difficult to calculate RMSDs. All RMSD calculations should be done using symmetry aware methods like RDKit's CalcRMS function or OpenBabel's obrms tool, which would not have a problem with changing atom orders. Although I doubt this will change the results much (other than to make them better), there's always the possibility that the DL algorithm is learning preferred atom orders in a way that favors a non-symmetry corrected, atom order dependent RMSD calculation. Furthermore, as I anticipate this to be a highly influential paper, it is important to stick with best practices (and not correcting for symmetry is definitely not best practice when reporting a docking RMSD). Please redo your RMSD calculations using appropriate methods or clarify that you are already using them.

We appreciate your careful attention and completely agree with your point on the importance of using symmetry-aware methods for RMSD calculations. In fact, we have utilized a symmetry-aware method, specifically the symmrmsd function from the spyrmsd package (<https://github.com/RMeli/spyrmsd>), for all RMSD computations in our work. We have added “we use a symmetry-aware method, specifically the symmrmsd function from the spyrmsd package [59] for all RMSD computations” to the main text.

You are absolutely correct that atom order should not influence the RMSD calculations. Our initial decision to use the PDBQT format over the SDF format was made prior to adopting symmetry-aware RMSD calculations. Recognizing the potential for misunderstanding, we have revised the text to remove any misleading statements. The updated text now reads, “The ligand input files for GNINA are in PDBQT format, created using OpenBabel after adding hydrogens with RDKit.”

Furthermore, we would like to assure you that the atom order does not impact the outcomes from our model. Since the small molecules are represented as graphs in our approach, where SMILES notation is employed solely to define bond and atom types, variations in atom orders do not influence the featurization and information propagation processes.

We hope these clarifications adequately address your concerns, and we are grateful for your insights that have helped enhance the precision and clarity of our manuscript.

Reviewer #2 (Remarks to the Author):

The authors present the development and validation of a new diffusion-based neural network model for ligand docking to flexible proteins. The major innovation has been the utilization of a morphing function between known protein-ligand complex structures and AlphaFold structures instead of the usual use of Gaussian noise during training of diffusion models. The manuscript is quite interesting to read but I have a few major concerns that need to be addressed:

We appreciate that the reviewer finds our paper interesting and acknowledges our innovative utilization of a morphing scheme. We have addressed the specific concerns below.

1. The authors split the dataset by time. This will result in a large overlap of homologous or even identical proteins in training and test set. In particular, regarding the study on the conformational change of kinases: It looks impressive that the model can predict DFG-in and DFG-out conformations. However, one has to recognize that this transition is conserved among the kinase family. Since there are many kinase structures in their training set, the model has learned the pattern of this transition (and not the underlying physics). Thus, it is not really surprising that the model is able to test both conformations during docking. This, however, is no prove of generalizability of the model.

We agree with the reviewer that the DFG-in/out transition is a conserved mechanism within the kinase family. The section in question aims to illustrate DynamicBind's ability to modulate kinase conformations according to interacting ligands that were not present in the training set. We believe that this generalizability is particularly valuable for screening small molecule drug candidates when the specific protein conformation they bind to is unknown. Beyond the DFG transitions, Figure 4 showcases other, less common conformational changes captured by the model in the test set, reinforcing the notion that the model learns more than just a repetitive pattern. Our intention is to highlight the model's ability to predict changes in protein structure dictated by ligand interactions, alleviating the requirement of holo-like conformation for traditional docking.

In this context, the authors show an improvement over DiffDock (Figure 2). However, when we trained DiffDock on a dataset where training and test set proteins were split by maximum diversity (to check generalizability) DiffDock drops to below 10% success rate (<2A) demonstrating the existing bias in those type of diffusion models. As DynamicBind is based on the same architecture, it is very likely that also DynamicBind will drop significantly in performance when applying it to novel proteins. I would recommend a split which maximizes the difference of protein homology between training and test set.

We appreciate the reviewer's recommendation to assess the generalizability of our model across proteins with diverse homology. In response, we've incorporated an expanded evaluation in Section 5.3 of the Supplementary Information, applying maximum sequence similarity cutoffs of 30%, 60%, and 90% between the training and test sets. While there was an observed decrease in model accuracy, it is encouraging that DynamicBind outperformed DiffDock at different similarity thresholds for the PDBbind and MDT test sets. This suggests a promising generalization capability, which we anticipate will be bolstered as more diverse experimental data becomes available for training. We have updated the main text to reflect these changes. In Section 2.2, we now state, "Results for individual datasets, as well as those filtered at 30%, 60%, and 90% maximum sequence similarity cutoffs, are detailed in Sections 5.2 and 5.3 of the Supplementary Information." Additionally, in the discussion section, we have revised the text to

include, “there is potential for further refinement, such as enhancing its generalization capabilities for proteins with low homology to the training set.”

2. In the Methods section the authors discuss the rigid and flexible version of Vina. In the Results section, however, I could only see one results column for Vina. Which version was used?

We used the rigid version of Vina for our primary analysis. Extra runs were also performed in Flex mode, with results added in Section 5.4 of the Supplementary Information, which were found to be comparable to those of the rigid mode (even slightly worse in the case of PDBbind test set). It's important to note that the Flex mode requires pre-defining which residues are flexible prior to docking. Flexible residues were chosen as those with any side chain atom within 5Å of the native ligand atoms. We opted not to include Flex mode results in the main text primarily because this mode, which requires pre-defining flexible residues, is less commonly used in typical docking protocols. The revised manuscript now states: “For Vina Flex mode, flexible side-chains must be predetermined. We identified all residues with side chain atoms within 5Å of the ligand atoms as flexible. In this mode, the protein backbone remains rigid.” As a side note, we encountered 'numerical error' crashes in 75 instances when attempting to use Gnina in flex mode. This may suggest a need for further refinement in its implementation, potentially attributable to the infrequent application of this mode in standard protocols.

3. I do not think that the comparison with Vina, Glide and Smina is fair if they are run in the “rigid protein” mode. Glide, for example, offers a validated induced fit modus. Why did the authors not use this “flexible protein” modus for a fairer comparison with DynamicBind?

We conducted Glide Induced-Fit Docking (IFD) as recommended and found that it only slightly outperforms standard Glide. This modest improvement is likely due to Glide IFD being more suited for protein side-chain movements rather than significant backbone motions, as noted in the Glide Induced Fit paper (miller2021reliable, JCTC): 'System 4 included cases involving significant backbone motion outside the scope of the present work.' Also, it is pertinent to point out the considerable computational resources required for Glide IFD, which entailed several weeks to fully complete the entire test set, while our methods can finish within a day. Detailed settings for Glide IFD and its results are included in the Methods section and Section 5.4 of the Supplementary Information, respectively.

4. It is not clear if the classical docking methods were run with the docking box covering the full protein which typically is not successful. The more robust method for blind docking with those classical docking methods is to first predict all possible binding sites and then perform docking to all sites in sequence, combine the poses and rank them according to score. Otherwise, I again do not believe that the current comparison is fair.

We completely agree that using an excessively large docking box is not a suitable practice for classical docking methods. Consequently, rather than utilizing global docking on the AlphaFold-predicted protein structure, we used local docking for these traditional methods. The docking

box was set with a 4Å buffer around the native ligand on all sides, in line with prior research \cite{corso2023diffdock}. Given concerns over computational costs, we provided the native binding site information to these classical methods. This approach confers them a minor advantage over deep learning-based methods, which conduct global docking without prior knowledge of the binding pocket's location. To clarify the adoption of local docking, the main text now reads “The docking box was defined using an automatic box around the native ligand with the default buffer of 4Å on all 6 sides. And the box center was the center of mass of the native ligand.”

5. Line 202: The authors state “In traditional docking methods, the sampling of protein conformations is decoupled from the docking step”. This is not correct as methods such as Glide and Vina-flexible sample ligand and protein conformations (at least on some level) at the same time.

We agree with the reviewer that the original statement is not accurate. Therefore, the text has been modified to “Conventional docking protocols usually perform protein conformation sampling as a separate step from the docking process [15,38].”

6. In chapter 2.5. the authors nicely show the potential of DynamicBind to identify ligands binding to cryptic sites. However, they only show a single result. How reproducible is this result for other systems? I would recommend that the authors select a few more PDB structures not homologous to any training set system from the Cryptosite website: <https://modbase.compbio.ucsf.edu/cryptosite/> and repeat the predictions.

Thank you for the valuable suggestion. We have now included four additional cases from the Cryptosite dataset, specifically chosen for their low maximum sequence similarity to our training set (below 30%). The resulting ligand RMSDs are around 2Å, indicating that DynamicBind successfully identified the cryptic sites in these instances. We have updated the main text to direct readers to these cases detailed in the Supplementary Information.

7. The authors provided a GitHub link. But no code is currently available at this location.

Thanks for the interest. We have now uploaded the code to the GitHub. We have modified the text from “Demo, instructions, and codes for DynamicBind will be available at <https://github.com/luwei0917/DynamicBind>.” To “Demo, instructions, and codes for DynamicBind are available at <https://github.com/luwei0917/DynamicBind>.”

We trust that these explanations have sufficiently resolved your queries. Your valuable feedback has been instrumental in refining the accuracy and coherence of our manuscript.

Reviewers' Comments:

Reviewer #1:

Remarks to the Author:

The authors have addressed all my concerns and the result is a stronger paper I look forward to seeing published.

Reviewer #2:

Remarks to the Author:

I appreciate the response of the reviewers to my comments. I confer with the response of the authors and their adaptations of the manuscript except with their response to point 1 (second part):

>>

We appreciate the reviewer's recommendation to assess the generalizability of our model across proteins with diverse homology. In response, we've incorporated an expanded evaluation in Section 5.3 of the Supplementary Information, applying maximum sequence similarity cutoffs of 30%, 60%, and 90% between the training and test sets. While there was an observed decrease in model accuracy, it is encouraging that DynamicBind outperformed DiffDock at different similarity thresholds for the PDBbind and MDT test sets. This suggests a promising generalization capability, which we anticipate will be bolstered as more diverse experimental data becomes available for training. We have updated the main text to reflect these changes. In Section 2.2, we now state, "Results for individual datasets, as well as those filtered at 30%, 60%, and 90% maximum sequence similarity cutoffs, are detailed in Sections 5.2 and 5.3 of the Supplementary Information." Additionally, in the discussion section, we have revised the text to include, "there is potential for further refinement, such as enhancing its generalization capabilities for proteins with low homology to the training set."

<<

A critical analysis of the pros and cons of new diffusion model approaches for docking compared to classical methods needs to be included in the main text (not just in the SI). Otherwise the limitations of those new methods remain hidden to other researchers.

In general, for test sets with low similarity to the training set the results fall significantly short of classical methods (Figure 9). This is in agreement with a recent study from Noe and coworkers (<https://www.biorxiv.org/content/10.1101/2023.11.03.565471v1.full.pdf>).

Even with a 90% similarity threshold, DynamicBind performs worse and equal to Gnina and Glide, respectively, on the PDBBind test set. This is in stark contrast to what is shown in Figure 2 in the main text, which suggested that DynamicBind significantly outperforms classical docking methods. I find this biased selection of results very troublesome for the scientific discourse and highly recommend further adaptation of the main text.

Dear Nature Communications Editors—

Thank you for the opportunity to revise our manuscript entitled, “DynamicBind: Predicting ligand-specific protein-ligand complex structure with a deep equivariant generative model” (NCOMMS-23-39499-T). Thanks also to the reviewers for their careful reading of our work and their thoughtful comments. The main text and Supplementary Information have all been revised and extended accordingly. In particular, the Supplementary Information now includes additional new figures, analyses, and explanations that respond to the reviewers’ suggestions. We think that, as a result of answering the concerns of the reviewers, the manuscript has been substantially improved.

Our specific responses to the questions and concerns of each reviewer are given below. The original remarks of the reviewers are shown in black text and our responses are given in blue text. The corresponding changes in the manuscript and methods files have also been highlighted with blue text.

REVIEWER COMMENTS

Reviewer #1 (Remarks to the Author):

The authors have addressed all my concerns and the result is a stronger paper I look forward to seeing published.

Thank you for your positive feedback and for acknowledging our efforts in addressing your concerns. We greatly appreciate your support, and we share your excitement about the potential publication of our paper.

In this revision, we have made several additions based on the suggestions provided by reviewer 2. Firstly, we have included a new figure (Fig 9) that illustrates the results stratified by the maximum ligand similarity to the training set. This addition aims to demonstrate the model's capability to generalize to new ligands effectively. We have also expanded the main text to provide a more detailed analysis of the strengths and weaknesses of our deep learning model in comparison to classical methods. We believe that this enhancement contributes to a more thorough understanding of our approach and its relation to other works in this field.

Once again, we sincerely appreciate your valuable feedback and support throughout the review process. We believe that these revisions, made in response to reviewer 2's suggestions, have further strengthened the paper.

Reviewer #2 (Remarks to the Author):

I appreciate the response of the reviewers to my comments. I confer with the response of the authors and their adaptations of the manuscript except with their response to point 1 (second part):

>>

We appreciate the reviewer's recommendation to assess the generalizability of our model across proteins with diverse homology. In response, we've incorporated an expanded evaluation in Section 5.3 of the Supplementary Information, applying maximum sequence similarity cutoffs of 30%, 60%, and 90% between the training and test sets. While there was an observed decrease in model accuracy, it is encouraging that DynamicBind outperformed DiffDock at different similarity thresholds for the PDBbind and MDT test sets. This suggests a promising generalization capability, which we anticipate will be bolstered as more diverse experimental data becomes available for training. We have updated the main text to reflect these changes. In Section 2.2, we now state, "Results for individual datasets, as well as those filtered at 30%, 60%, and 90% maximum sequence similarity cutoffs, are detailed in Sections 5.2 and 5.3 of the Supplementary Information." Additionally, in the discussion section, we have revised the text to include, "there is potential for further refinement, such as enhancing its generalization capabilities for proteins with low homology to the training set."

<<

A critical analysis of the pros and cons of new diffusion model approaches for docking compared to classical methods needs to be included in the main text (not just in the SI). Otherwise the limitations of those new methods remain hidden to other researchers.

In general, for test sets with low similarity to the training set the results fall significantly short of classical methods (Figure 9). This is in agreement with a recent study from Noe and coworkers (<https://www.biorxiv.org/content/10.1101/2023.11.03.565471v1.full.pdf>).

Even with a 90% similarity threshold, DynamicBind performs worse and equal to Gnina and Glide, respectively, on the PDBBind test set. This is in stark contrast to what is shown in Figure 2 in the main text, which suggested that DynamicBind significantly outperforms classical docking methods. I find this biased selection of results very troublesome for the scientific discourse and highly recommend further adaptation of the main text.

Thank you for your insightful feedback. We fully agree that a detailed analysis of the pros and cons of our deep learning model in comparison to classical methods is essential in the main text. Consequently, we have made appropriate revisions in the section 2.2 and discussion section, which are detailed below.

Furthermore, we respectfully wish to address your concerns regarding the perceived bias in our results selection, offering clarification from three specific perspectives:

Firstly, regarding our test cases, it's important to note that while some proteins may be identical or homologous to those in previously deposited crystal structures, the ligands involved are often different. This variation in ligands is crucial, as newly deposited crystal structures typically present significant differences from prior ones to merit publication. In our test set, only 8.8% of the cases exhibit both protein and ligand similarities above 0.9 and 0.6, respectively, compared to the training set. It is common in drug discovery for the same protein to be studied due to its interaction with new ligands, which can induce diverse conformational changes. To highlight this aspect, we have included a new figure (Fig 9) that illustrates the model's

performance with varying degrees of ligand similarity. This demonstrates our model's effectiveness in scenarios where the protein is known, but the ligand is novel.

Secondly, the task of docking a new ligand remains challenging, even when the co-crystallized structure of a protein and its ligand is available. Classical methods, which are based on force-field models, are generally effective in placing co-crystallized ligands into the corresponding holo protein conformations. However, these methods, limited by high computational costs for full protein flexibility, typically assume a mostly rigid protein backbone. This assumption increases their sensitivity to small ligand changes, especially when a new ligand triggers considerable conformational changes in the protein. Our model, in contrast, is designed to better accommodate such dynamic interactions, offering a distinct advantage in situations where ligand-induced protein conformational changes are significant.

Thirdly, the comparatively better performance of classical methods under lower protein similarity thresholds can be partly attributed to their focus on local docking within pre-defined ground-truth pockets. Given that the identification of binding sites on new proteins is itself an active area of research, the challenge faced by deep learning methods in blind global docking on new proteins are likely common across various approaches.

In summary, to present a detailed analysis of the pros and cons of our method in comparison to classical methods, we have added a new paragraph to the main text under Section 2.2. It now reads "To assess the model's generalization to new proteins and ligands, we analyzed results stratified by maximum ligand and protein sequence similarity to the training set (Section 5.3) This analysis reveals that DynamicBind performs well with new ligands, outperforming others, but is less effective with new proteins, where it is outperformed by classical docking methods with pre-defined ground-truth binding pockets. Other deep learning methods also show similar declines with new proteins, hinting at a need for larger training set and improved inductive biases. Moreover, considering that the identification of binding sites on new proteins is an active research area, the challenges encountered in blind global docking by deep learning methods, including ours, are likely shared across different approaches. Overall, DynamicBind's proficiency with new ligands is significant in drug discovery, highlighting its potential in identifying protein conformational changes vital for creating effective, specific drugs." We have also revised a paragraph in the discussion section, now reads "DynamicBind, while demonstrating state-of-the-art performance in our benchmarks, still presents opportunities for improvement, especially in enhancing its ability to generalize to proteins with low sequence homology compared to those in the training set [47, 48]." In addition, a figure (Fig 9) has been added in section 5.3 showing the result stratified at different ligand similarity threshold.

We believe these changes substantially strengthen our paper and hope they adequately address your concerns. We appreciate the opportunity to enhance our work through your constructive feedback.

Fig. 9 Benchmark results with bootstrapped confidence interval for PDBbind and MDT test set filtered at different maximum ligand similarity to the training set. For each method, we resample 100 times with replacement, and plot 68% confidence intervals.